# Analysis of the effects of isolation piles on the basal stability of foundation pits against upheaval based on continuous velocity fields

Yi Zhou[1], Yu Shao[2], Shaokun Ma[1] *

**1** School of Civil Engineering and Architecture, Guangxi University, Nanning, Guangxi, China, **2** Guangxi Transportation Design Group Co., Guangxi, China

* mashaokun@gxu.edu.cn

**Data Availability Statement:** All relevant data are within the paper and its Supporting information files.

## Abstract

The upper bound limit analysis method is one of the main approaches to check the basal stability of foundation pits against upheaval. However, existing studies have often failed to consider the effects of external supporting structures, including isolation piles and others, on basal stability against upheaval. This study derives a formula for the coefficient of basal stability against upheaval under the action of isolation piles by simplifying the pile-soil relationship and systematically examines the impact of isolation pile parameters on basal stability against upheaval using the theory of continuous velocity fields and the upper bound limit analysis method. A comparison of simulation results indicates that this technique can accurately identify the variation trend of basal stability against upheaval under the influence of isolation piles and achieve high calculation precision under the operating conditions of wide foundation pits and short isolation piles. Accordingly, a moderate increase in isolation pile parameters produces a significant supporting effect for narrowed foundation pits. Whereas for wide foundation pits, the supporting capacity of isolation piles can be maximized when pile length equals excavation depth.

## 1. Introduction

In engineering practice, external supporting structures are constructed to reduce the deformation of foundation pits and adjacent buildings. In this regard, isolation piles are the primary external supporting structures due to their mature construction technologies and well-proven support capacity [1].

With various supporting structures, verifying basal stability against upheaval becomes an essential task of foundation pit design, which can comprehensively represent the stability of foundation pits. Investigating basal stability against upheaval is relatively mature, and the primary approaches include the finite element method [2,3], the limit equilibrium method [4,5], and the limit analysis method [6].

The upper bound limit analysis method can obtain an accurate upper bound of the ultimate load by constructing a deformation mechanism with a kinematically admissible velocity field

**Funding:** The project is funded by the National Natural Science Foundation of China (No.52268062), Guangxi Natural Science Foundation Key Project (No.2020GXNSFDA238024), Key Science and Technology Projects in Guangxi's Transport industry (No.GXJT-2020-02-08), Guangxi Key R&D Program Project (No.GUIKE2021AB22177).

**Competing interests:** The authors have declared that no competing interests exist.

and establishing an equation of virtual work in which internal energy dissipation equals external work. The key factor affecting the calculation precision of the limit analysis method is the construction of a deformation mechanism with a kinematically admissible velocity field [7].

Common kinematic mechanisms, such as Terzaghi mechanisms, Prandtl mechanisms [8], Hill mechanisms [9], and multi-block kinematic mechanisms [10,11], are often composed of rigid blocks and uniform or non-uniform deformation zones in existing studies. The above investigations have adopted the rigid-plastic hypothesis, which assumes that a deformation mechanism contains many velocity discontinuities. However, soil deformation appears to be continuous in a real foundation pit.

In view of this, scholars have investigated deformation mechanisms as continuous velocity fields. Klar [12] determined the *p-y* curves of piles subjected to lateral loads using the upper bound method based on continuous deformation mechanisms. Tang et al. [7] developed a continuous velocity field model applicable to soft soil layers and calculated the safety coefficient of basal stability against upheaval for different types of foundation pits. The continuous velocity field model was found to be more consistent with reality than other kinematic mechanisms.

However, the above analysis method frequently disregards the effects of external supporting structures, including isolation piles, on the basal stability of foundation pits against upheaval. Many studies have demonstrated that isolation piles can effectively limit soil displacement and constrain structural deformation [13,14]. However, few theoretical studies have investigated the ability to isolate piles to constrain basal upheaval.

Based on the continuous velocity field model suggested by Tang et al. [7], this study proposes a method for calculating the basal stability of foundation pits against upheaval considering the supporting capacity of isolation piles. This method simplifies the pile-soil interaction relationship and obtains reference-worthy calculation results through numerical integration. Besides this, we compare the calculation results with numerical simulations and conduct an error analysis to identify the specific effects of isolation piles.

## 2. Basic principles of the upper bound limit analysis method

Based on the upper bound theorem [9], the internal energy dissipation rate in any deformation mechanism with a kinematically admissible velocity field cannot be less than the external work rate. Hence, the determined external load must be greater than or equal to the actual ultimate load, as expressed in the equation below:

$$\int_S T_i v_i \mathrm{d}S + \int_V X_i v_i \mathrm{d}V \le \int_V \sigma_{ij} \dot{\varepsilon}_{ij} \mathrm{d}V \qquad i, j = 1,\ 2,\ 3. \tag{1}$$

where $\ddot{\varepsilon}_{ij}$ is plastic strain rate; $v_i$ is a velocity field in geometric compatibility with $\ddot{\varepsilon}_{ij}$; $T_i$ and $X_i$ are the surface force vector at boundary $S$ and the physical force vector in region $V$, respectively; $\sigma_{ij}$ is a stress field associated with $\ddot{\varepsilon}_{ij}$.

For the Tresca yield criterion, the demonstrations and applications by Shield, Tang et al. [7,15] indicated that Eq (1) can be written as follows:

$$\int_S T_i v_i \mathrm{d}S + \int_V X_i v_i \mathrm{d}V \le \int_V 2s_u \left| \dot{\varepsilon}_{max} \right| \mathrm{d}V \tag{2}$$

where $s_u$ is yield strength; $\dot{\varepsilon}_{max} = \frac{1}{2}\dot{\gamma}_{max}$ and $\dot{\gamma}_{max}$ is the maximum engineering shear strain rate.

## 3. Construction of compatible velocity fields

In the general calculation process of the upper bound limit analysis method, the above energy equation for problem solving must be applied to the constructed deformation mechanism with a kinematically admissible velocity field to determine the upper bound of the ultimate load. In this paper, the deformation mechanism is constructed utilizing the continuous velocity field model proposed by Tang et al. [7], as expressed below.

Fig 1 reveals the basic assumptions of the deformation mechanism, which are as follows: the retaining wall itself does not deform; the degree of soil deformation behind the wall is consistent with that of surface settlement, and the dotted lines in Fig 1 show its movement trajectory; the deformation distribution of soil does not change in the vertical trajectory direction; the soil inside the deformation mechanism experiences uniform deformation, while that outside it does not deform.

Considering that soil deformation depends on surface settlement and that the surface settlement curves of soil vary with specific geological conditions, this paper uses a surface settlement curve form applicable to soft soil layers [7,16] as follows:

$$v_y = \frac{\beta v_m x}{l} \left\{ e^{(-\frac{2x}{l})} - e^{(-\alpha)} \right\} \tag{3}$$

where $l = \alpha H$ is the range of surface settlement behind the retaining wall; $x$ is the horizontal distance between soil and the retaining wall; $H$ is foundation pit excavation depth; $\beta$ is the normalization coefficient, which makes the calculated maximum surface settlement equal to the real one; $\alpha$ is influence coefficient.

The value of $\alpha$ is related to $H/B$. In order to obtain a smaller upper bound in subsequent calculations, the value of $\alpha$ should be taken with reference to Table 1, according to the findings of Tang et al [7]. In the table, B denotes the width of the foundation pit.

The above mechanism with the velocity field is divided into three areas: *abof*, *boc*, and *cdeo*. The velocity field expressions of regions *abof*, *boc*, and *cdeo* can be obtained from Eq (3) based on boundary and deformation compatibility conditions. Specifically, regions *abof* and *cdeo*

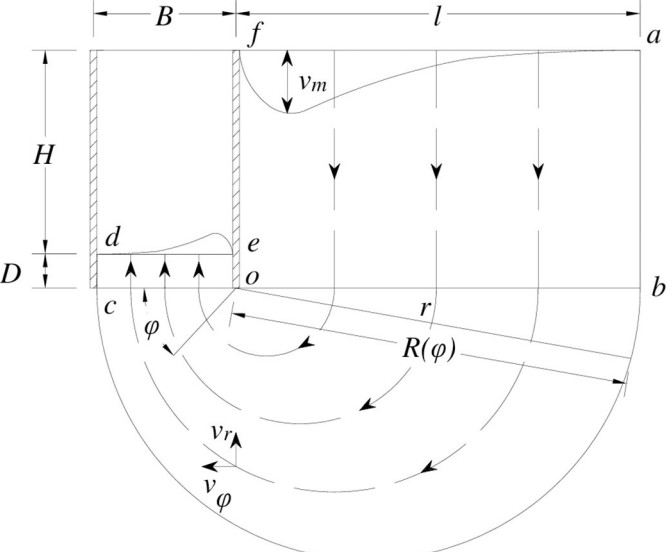

**Fig 1. Deformation mechanism of foundation pit.**

**Table 1. Recommended values for $\alpha$.**

| H/B | $\alpha$ | H/B | $\alpha$ |
|---|---|---|---|
| 0.25 | 6.0 | 1.50 | 2.5 |
| 0.50 | 4.5 | 2.00 | 2.1 |
| 0.75 | 4.0 | 3.00 | 1.6 |
| 1.00 | 3.5 | 4.00 | 1.5 |

adopt a rectangular coordinate system and consider the soil flow direction as positive. Region *boc* adopts a polar coordinate system, which is specified to take the counterclockwise direction as positive.

Region *abof*:

$$v_{\text{x-abof}} = 0 \tag{4}$$

$$v_{\text{y-abof}} = \frac{\beta x}{l}\left\{e^{\left(-\frac{\alpha x}{l}\right)} - e^{(-\alpha)}\right\} \tag{5}$$

where $v_{\text{x}}$ is the velocity in the horizontal direction; $v_{\text{y}}$ is the velocity in the vertical direction.

Region *boc*:

$$v_{\varphi\text{-boc}} = -\frac{l\beta r}{R(\varphi)^2}\left\{e^{\left(-\frac{\alpha r}{R(\varphi)}\right)} - e^{(-\alpha)}\right\} \tag{6}$$

$$v_{\text{r-boc}} = -\frac{l\beta r}{R(\varphi)^3}\frac{\partial R(\varphi)}{\partial \varphi}\left\{e^{\left(-\frac{\alpha r}{R(\varphi)}\right)} - e^{(-\alpha)}\right\} \tag{7}$$

where $v_{\varphi}$ is the velocity in velocity in the direction perpendicular to the polar radius direction; $v_{\text{r}}$ is the velocity in the polar radius direction; $r$ is the distance between a soil element and pole shank $o$; $R(\varphi)$ is the polar radius, defined as:

$$R(\varphi) = \sqrt{a_1^2 + a_2^2 - 2a_1 a_2 \cos(\varphi - \arcsin(\frac{a_2}{a_1}\sin(\varphi)))} \tag{8}$$

For the sake of obtaining the minimum upper bound solution [7] it has $a_1 = \frac{l+B}{2}$ and $a_2 = \frac{l-B}{2}$ when $H/B > 0.25$.

Region *cdeo*:

$$v_{\text{x-cdeo}} = 0 \tag{9}$$

$$v_{\text{y-cdeo}} = \frac{l\beta x}{B^2}\left\{e^{\left(-\frac{\alpha x}{B}\right)} - e^{(-\alpha)}\right\} \tag{10}$$

## 4. Calculation of the safety coefficient

The safety coefficient is defined in various forms by different basal stability analysis methods [17]. Based on Eq (2) given by Tang et al. [7] then:

$$F_s = \frac{\sum \int_V 2s_u |\dot{\varepsilon}_{max}| dV}{\sum \int_V \gamma_{sat} v_y dV} \tag{11}$$

where $F_s$ is the safety coefficient; $\gamma_{sat}$ is soil saturated unit weight; $s_u$ is soil strength; $v_y$ is the vertical velocity component in the mechanism with the velocity field; $\dot{\varepsilon}_{max}$ is engineering shear strain, which can be calculated in different coordinate systems as follows:

$$\text{Rectangular coordinate system}: \dot{\varepsilon}_{max} = \frac{1}{2}\left(\frac{\partial v_x}{\partial y} + \frac{\partial v_y}{\partial x}\right) \tag{12}$$

$$\text{Polar coordinate system}: \dot{\varepsilon}_{max} = \sqrt{\left(\frac{\partial v_r}{\partial r}\right)^2 + \frac{1}{4}\left(\frac{1}{r}\frac{\partial v_r}{\partial \varphi} + \frac{\partial v_\varphi}{\partial r} - \frac{v_\varphi}{r}\right)^2} \tag{13}$$

Since Eq (11) considers only the work done by the gravity of soil, the following assumptions are made to calculate the effects of isolation piles on soil in the above kinematic mechanism: Isolation piles are not displaced due to soil movement, and their dead weight can be ignored; isolation piles, assumed to be rigid bodies, do not deform under soil pressure; the isolation piles are independent of the enclosure structure without affecting each other; the soil arching effect generated when pile spacing is at the allowable clear span is disregarded [18]; the presence of isolation piles does not change the original form of the kinematic mechanism.

Under the above assumptions, isolation piles come into contact with moving soil under pressure and produce a frictional resistance to soil. At this time, the kinematic mechanism retains its form described in the previous section. Under the action of isolation piles, Eq (11) is rewritten as follows according to the *Technical code for building pile foundations* (JGJ94-2008) [19]:

$$F_s = \frac{\sum \int_V 2s_u |\dot{\varepsilon}_{max}| dV}{\sum \int_V \gamma_{sat} v_y dV + \sum \int_V Q_s v dV} \tag{14}$$

where $v$ is the total velocity of various calculation regions; $Q_s$ is the standard value of the negative frictional resistance on the pile side. $Q_s$ is calculated as follows:

$$Q_{si} = -\zeta_i \sigma_i \tag{15}$$

$$\sigma_i = P + \sigma_{ri} \tag{16}$$

$$\sigma_{ri} = \sum_{i=1}^{j} \gamma_i \Delta z_i \tag{17}$$

where $i$ is the $i^{th}$ layer of soil with a range of $1-j$; $Q_{si}$ is the standard value of the negative frictional resistance on the pile side in the $i^{th}$ layer of soil; $\zeta_i$ is the negative frictional resistance coefficient of the $i^{th}$ layer of soil, and the value is provided in Table 2 [19]; $\sigma_i$ is the average vertical effective stress of the $i^{th}$ layer of soil; $\sigma_{ri}$ is the average vertical effective stress of the $i^{th}$ layer of soil generated by the dead weight of soil; $\gamma_i$ is the unit weight of the $i^{th}$ layer of soil, set

**Table 2. Negative frictional resistance coefficient $\zeta_i$.**

| Types of soil | $\zeta_i$ |
|---|---|
| Saturated soft soil | 0.15–0.25 |
| Clayey soil, powdered soil | 0.25–0.40 |
| Sandy soil | 0.35–0.50 |
| Self-weighted wet sinking loess | 0.20–0.35 |

as the buoyant unit weight when the soil layer is located below the groundwater level; P is surface uniformly distributed load; $\Delta z_i$ is the thickness of the $i^{th}$ layer of soil.

In Eq (14), the range of integration is related to pile length $t$ and pile-pit spacing $e$, so calculation related to isolation piles will show the following two forms:

When pile length is less than foundation pit excavation depth, it is necessary to consider only the work done by the isolation piles in the region *abof*, as stated below:

$$\sum \int_V Q_s v \mathrm{d}V = \frac{2r}{s} \int_0^t Q_s v_{\text{y-afob}} \mathrm{d}y \tag{18}$$

where $r$ is isolation pile radius; $s$ is pile spacing; $t$ is pile length.

In this paper, the three-dimensional problem is simplified into a two-dimensional one for the convenience of calculation. The following examples characterize isolation piles' effects on basal stability's safety coefficient against upheaval within a unit width. Therefore, in Eq (18), $\frac{2r}{s}$ is the number of isolation piles within a unit width.

When pile length $t$ is greater than foundation pit excavation depth $H$, it is necessary to calculate the work of the isolation piles in region *boc*, as shown below:

$$\sum \int_V Q_s v \mathrm{d}V = \frac{2r}{s} \left( \int_0^H Q_s v_{\text{y-abof}} \mathrm{d}y + \int_{\pi - \arctan(\frac{t-H}{e})}^{\pi} Q_s v_{\text{boc}} \varphi \mathrm{d}\varphi \right) \tag{19}$$

where $e$ is pile-pit spacing; $v_{\text{boc}} = \sqrt{v_{\text{phi-boc}}^2 + v_{\text{r-boc}}^2}$ is the total velocity of soil movement in region *boc*.

In the assumptions, this paper only considers the work done by frictional resistance on the deformation mechanism during pile-soil contact and ignores the effect of isolation piles on the form of velocity fields. In engineering practice, the pile-soil relationship is often complex. Many studies and experiments [20,21] have revealed that when pile spacing is small enough, a soil arching effect will occur between soil and piles, making it possible for piles to do more negative work on the soil.

In existing studies, it is difficult to define a reasonable expression of an arch axis consistent with the real situation [18], making it difficult to quantify the work done by isolation piles on soil under the soil arching effect. For this reason, this paper ignores the extra negative work generated due to the soil arching impact in region *boc* and its effect on the deformation mechanism. A smaller upper bound solution will be obtained under the above assumptions, which means that the safety coefficient of basal stability against upheaval obtained by this method is relatively conservative and safe.

## 5. Numerical simulation validation

This paper compares numerical simulation and theoretical calculation results in the subsequent calculation of the coefficient of basal stability against upheaval under different operating

conditions to validate the accuracy of this algorithm. For specific work, we have used the finite element software Plaxis, the mathematical software MATLAB and Mathematica.

Ugai [22] and Matsui [23] used finite element analysis with reduced shear strength to analyze slope stability in two-dimensional scenarios. Similarly, Faheem et al. [2] applied this approach to the analysis of basal stability against upheaval and obtained relatively accurate results.

These studies induce the failure of nonlinear analysis by gradually reducing the shear strength parameters of soil to determine the safety coefficient of basal stability against upheaval $N$ [2]. Specifically, the reduced shear strength parameters $c'_N$ and $\varphi'_N$ are substituted for the corresponding parameters $c'$ and $\varphi'$ in the shear strength equation $\tau_f = c' + \sigma' \tan \varphi'$, where $c'_N = c'/N$, $\varphi'_N = \tan^{-1}(\tan(\varphi')/N)$. Ultimately, the shear strength equation $\tau_{fN} = c'_N + \sigma' \tan \varphi'_N$ is obtained after stiffness reduction. When the value of $N$ is small enough, finite element calculation results tend to be almost elastic. Since the calculated step size is mainly controlled by $N$, if there is still no convergence after a certain number of iterations, the $N$ before this step size will be adopted as the only safety coefficient $N_c$. This method is referred to as the shear strength reduction finite element method (SSRFEM).

In this paper, isolation piles are added to the examples adopted by Tang et al. [7] and Faheem [2], as detailed below:

Foundation pit excavation depth $H$ = 9m; soil strength $s_u$ = 35kPa; unit weight $\gamma$ = 20kN/m$^3$; retaining wall's embedded depth in soil $D$ = 0; isolation pile radius $r$ = 0.8m; pile spacing $s$ = 1m. The specific values of pile length $t$ and pile-pit spacing $e$ are given in the following section. It is assumed in the examples that a soft soil layer surrounds foundation pits. Based on Table 2, $\zeta_i$ is set at 0.25 to obtain a smaller upper bound solution. In engineering practice verification, the value of $\zeta_i$ should be selected from the range given in Table 2 based on the needs of safety reserve. The relationship between the stability coefficient $N_c$ and the safety coefficient of basal stability against upheaval $F_s$ can be expressed as follows:

$$N_c = F_s \gamma_{sat} H / s_u \tag{20}$$

## 6. Error analysis based on the upper bound method

By comparing the results of upper bound analysis and numerical simulation under different working conditions in the above examples, the precision of upper bound analysis is related to multiple parameters. For a more accurate studying how calculation precision is related to $H/B$, pile length $t$, and pile-pit spacing $e$ (expressed as a multiple of excavation depth $H$), when $H/B$ = 1.0, the relationships of $N_c$ with $e$ and $t$ are shown in Fig 2.

Fig 2 indicates that the variation trend of the stability coefficient calculated by upper bound analysis is consistent with the results obtained by SSRFEM, even when isolation pile parameters are designed differently. And the error values between the theoretical calculation and the results of SSRFEM is small. This suggests that the upper bound analysis method proposed in this paper can accurately describe the trend of how the change in isolation pile parameters affects the coefficient of basal stability against upheaval in the case of a wide foundation pit.

By comparing values of the stability coefficient under different pile lengths and pile-pit spacings, it is evident that the increase in pile length is an essential factor affecting the precision of upper bound analysis. When the pile length exceeds the excavation depth, the calculation error of the stability coefficient gradually increases when pile-pit spacing remains unchanged.

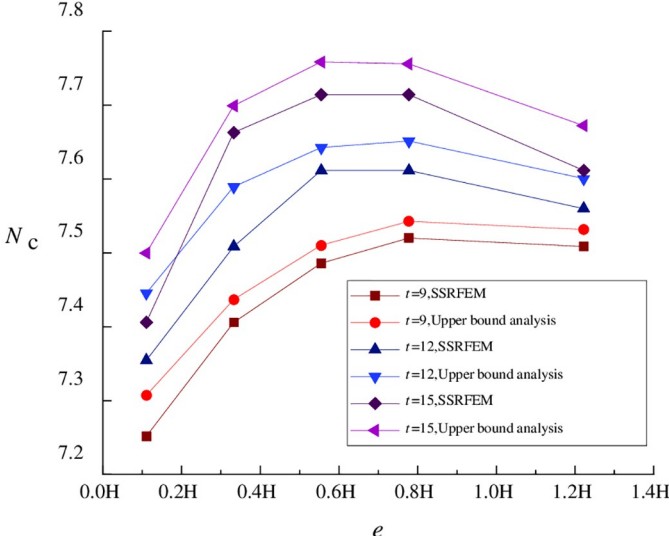

**Fig 2. Relationships of $N_c$ with $e$ and $t$ ($H/B = 1.0$).**

In the continuous velocity field adopted in this paper, the soil movement direction in region *afob* is parallel to the pile direction. Thus, when pile length is less than excavation depth, the primary error source is the effects of piles on soil displacement. Soil movement in the real situation will be constrained by the action of isolation piles, causing the upper bound solution in the theoretical limit state to be greater than the stability coefficient obtained by SSRFEM. At the same time, when some isolation piles are located in region *boc*, the part beyond excavation depth $H$ will experience a soil arching effect, which will subject soil movement to more restrictions and cause the calculation error of long piles greater than that of short piles. With increasing pile length, the above errors will gradually accumulate, resulting in lower calculation precision in the case of long piles.

In contrast, for a wide foundation pit, when isolation piles are 0.3 $H$–0.8 $H$ away from (instead of being closer to or farther away from) the foundation pit, the upper bound analysis will be more precise, with an error of about 1.0%. Based on Eqs (3) and (14), this is a region where the maximum surface settlement often occurs. On the other hand, this phenomenon mainly depends on the width-to-depth ratio of the foundation pit, as detailed in the follow-up analysis.

For a narrow foundation pit, when $H/B = 4.0$, the relationships of $N_c$ with $e$ and $t$ are shown in Fig 3. Apparently, for a narrow foundation pit, upper bound analysis is equivalent to SSRFEM in terms of the variation trend of the stability coefficient but has a significantly reduced overall calculation precision. The occurrence of the above phenomenon is attributable to different width-to-depth ratios.

Based on Eq (3) and the conclusion of Tang et al. [7], the width-depth ratio is directly proportional to basal stability against upheaval. This means that, in the case of narrow foundation pits, surface settlement is smaller, soil deformation is less obvious, and it is more difficult for actual soil deformation to approach the limit state. As a result, the error of the upper bound solution obtained by limit analysis is greater. In particular, when isolation piles are located in a region with a relatively large surface settlement, the calculation error reaches about 10%. A comparison between Figs 2 and 3 shows that the foundation pit's width-depth ratio is the primary factor affecting the precision of upper bound analysis.

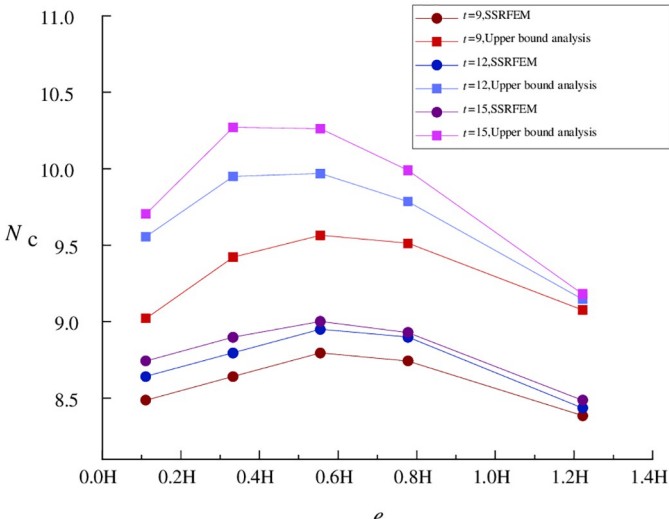

**Fig 3. Relationships of $N_c$ with $e$ and $t$ ($H/B$ = 4.0).**

Similar to wide foundation pits, increasing pile length also worsens the precision of upper bound analysis. However, pile length is only a secondary factor affecting the precision of upper bound analysis relative to the width-depth ratio. When isolation piles are far enough from the foundation pit, there is no apparent difference between wide and narrow foundation pits in calculation precision (error) under various pile lengths because the soil velocity in the theoretical calculation formula is relatively low.

However, the difference between wide and narrow foundation pits is that pile-pit spacing has a smaller effect on the precision of upper bound analysis for narrow foundation pits. This is because the larger soil displacement in the case of wide foundation pits is closer to the limit state assumed by upper bound analysis. In the same scenario where isolation piles are located in an area with a relatively large surface settlement, the calculation precision of wide foundation pits is higher and more sensitive to changes in isolation pile parameters. In the case of narrow foundation pits, the change in pile-pit spacing does not significantly affect calculation errors.

## 7. Effect analysis of isolation piles

### 7.1 Effect of pile length

A reasonable design of pile length can reduce engineering costs while minimizing basal upheaval in engineering practice. By keeping pile-pit spacing $e$ constant, this section discusses the variation in the safety coefficient of basal stability against upheaval for various pile lengths.

Eq (17) demonstrates that when pile length $t$ is less than excavation depth, the influence coefficient is positively correlated with pile length. Therefore, the following calculations only consider the impact of the increase in pile length $t$ when the pile length is $t \geq H$ ($H = 9m$). When $e$ = 1m and $e$ = 2m are obtained by theoretical calculation, the relationship between pile length $t$ (expressed as a multiple of excavation depth $H$) and stability coefficient $N_c$ is shown in Figs 4 and 5.

Figs 4 and 5 reveal that under different pile-pit spacings, the degree of the effect of pile length on stability coefficient $N_c$ depends mainly on the magnitude of $H/B$. When the foundation pit is wider, the impact of the change in pile length $t$ on the stability coefficient is smaller.

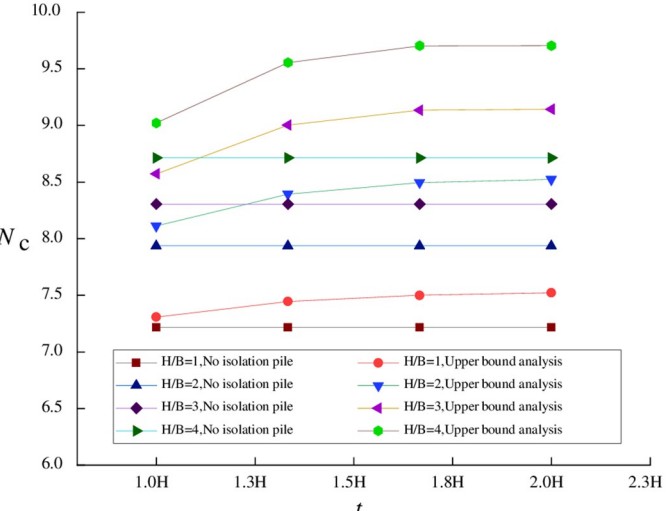

**Fig 4. Relationship between stability coefficient and *t* (*e* = 1m).**

In general, the stability coefficient increases as the pile length rises. The stability coefficient becomes stable when the pile length reaches around 1.8 H. This phenomenon is more apparent when isolation piles are closer to the foundation pit.

Similarly, when $H/B \leq 2$, the relationship between pile length and the influence coefficient approaches linearity. This is because the width-depth ratio of the foundation pit affects the value of the influence coefficient of the range of surface settlement ($\alpha$) in Eq (3). In other words, the larger the foundation pit's width, the greater the surface settlement range. Under a specific pile-pit spacing, the overall $t$-$N_c$ relationship of a wide foundation pit is closer to the first half of the $t$-$N_c$ relationship of a narrow foundation pit.

To sum up, when $H/B$ is relatively large, improving the parameters of the isolation pile can effectively increase the basal stability of foundation pits against upheaval. For a foundation pit

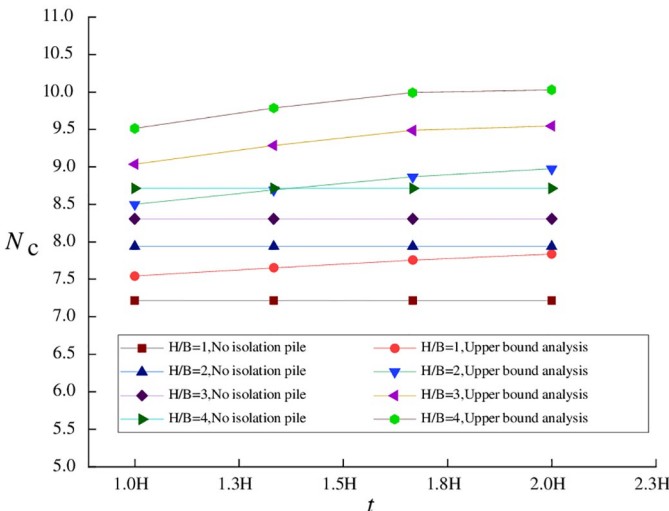

**Fig 5. Relationship between stability coefficient and *t* (*e* = 7m).**

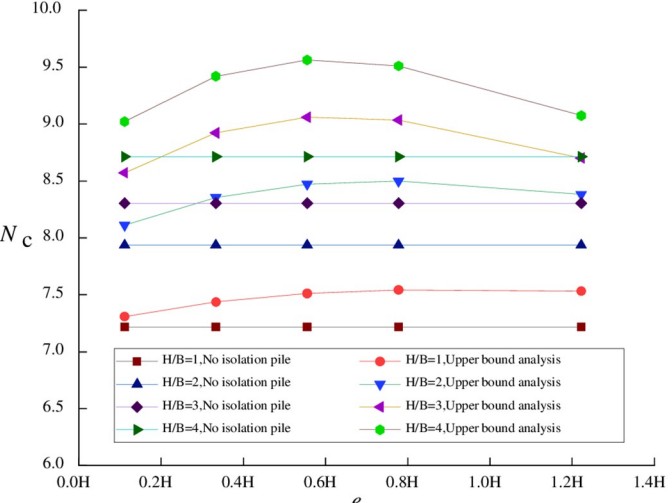

**Fig 6. Relationship between the stability coefficient and $e$ ($t$ = 9m).**

with a relatively small $H/B$, a pile length equal to the foundation pit excavation depth is sufficient to produce a satisfactory supporting effect, and the benefits of increasing pile length are minimal.

## 7.2 Effect of pile-pit spacing

In order to study the effect of pile-pit spacing on basal stability against upheaval, the pile length $t$ is held constant in this section. Figs 6 and 7 illustrate the relationship between stability coefficient $N_c$ and e (expressed as a multiple of excavation depth $H$) under various operating conditions.

When the foundation pit is relatively narrow ($H/B \geq 3$)due to the relatively small soil deformation, the effect of isolation piles on improving basal stability against upheaval is so limited that increasing pile-pit spacing to 0.8 $H$ does not impact the stability coefficient. Meanwhile, similar to pile length, the width-depth ratio of the foundation pit is also the main factor affecting the stability coefficient.

For a wide foundation pit, the maximum stability coefficient can be obtained when the value of $e$ is around the extreme point of Eq (5). At this point, isolation piles are often located in the region with the maximum surface settlement at the pit edge. This phenomenon becomes more evident when $H/B$ is larger.

To confirm the applicability of the above rule to long piles, when pile length exceeds excavation depth ($t$ = 15m), the relationship between stability coefficient and $e$ can be depicted, as revealed in Fig 7.

Fig 7 shows that the above rule still applies to long piles, although the extreme point varies depending on width-depth ratios. When pile length $t$ exceeds excavation depth, the calculation of the stability coefficient no longer depends solely on Eq (5) but requires more complex polar-coordinate integration operations.

Therefore, when short piles ($t \leq H$) are used in engineering practice, the maximum basal stability coefficient can be obtained by calculating the extreme point of the surface settlement curve and placing isolation piles near the extreme point. When it comes to long piles ($t > H$), numerical integration is required to be performed by Eq (11).

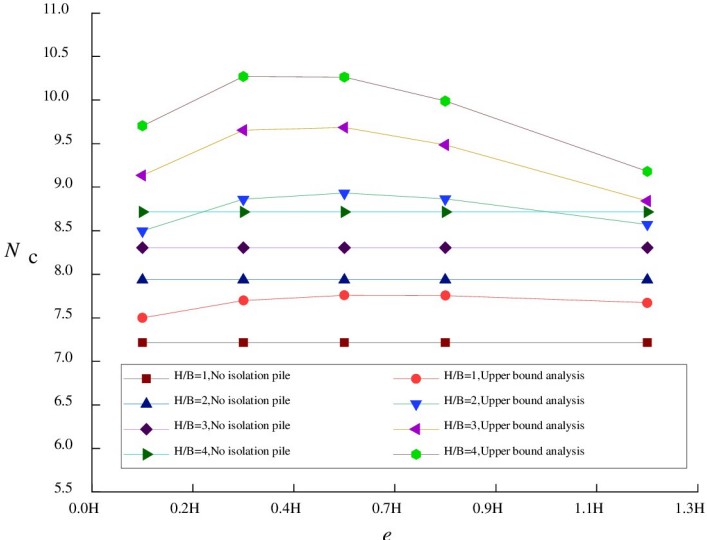

**Fig 7. Relationship between the stability coefficient and *e* (*t* = 15m).**

## 8. Discussion and conclusion

This study derives the upper bound analytical solution of basal stability against upheaval under the action of isolation piles based on deformation mechanisms in the form of continuous velocity fields. It simplifies the impact of supporting structures from a three-dimensional problem to a two-dimensional one. A comparison with SSRFEM results validates the accuracy and applicability of the method proposed in this paper. The calculation reveals that this method adheres to the following rules in terms of calculation precision:

1. For a foundation pit with any width-to-depth ratio, this method accurately identifies the variation trend of the stability coefficient under the various designed isolation pile parameters. This enables researchers always to obtain the isolation pile parameters that give full play to the optimal supporting effect.

2. The width-to-depth ratio of the foundation pit is the primary factor affecting the precision of upper bound analysis. The method proposed in this paper has a high calculation precision in the case of wide foundation pits, primarily when isolation piles are located in a region with relatively large surface settlements.

3. Pile length and pile-pit spacing play secondary roles in affecting calculation precision. The method proposed in this paper is more suitable for calculating the impact of isolation piles on basal upheaval when pile length is less than excavation depth. Pile-pit spacing has a particular effect on calculation precision in the case of wide foundation pits, whereas it has little influence in the case of narrow foundation pits.

Utilizing the above methods, this paper determines the effect of isolation piles on basal stability against upheaval under various operating conditions, specifically:

1. When the pile length exceeds the foundation pit excavation depth, the degree of the pile length's effect on the stability coefficient mainly depends on the width-depth ratio of the foundation pit. Compared with wide foundation pits, narrow foundation pits are more sensitive to the change in supporting structure parameters in terms of basal stability against

upheaval, which produces great benefits. A satisfactory supporting effect can be achieved in wide foundation pits when the pile length is equal to the foundation pit's excavation depth.

2. For short piles (pile length less than foundation pit excavation depth), locating isolation piles near the region with the maximum surface settlement often achieves the maximum coefficient of basal stability against upheaval. For long piles (pile length greater than foundation pit excavation depth), the extreme point of the $e$-$N_c$ relationship can be obtained only through the complex integration operations introduced in this paper.

Finally, the paper presents a series of assumptions relating to the interaction of the isolation piles with the soil. Moreover, the deformation mechanism adopts an approximate form of the displacement's elastic solution. As a result, the method proposed in this paper deviates somewhat from the actual soil deformation and fails to be applied to complex engineering conditions.

Although we are currently unable to specify the exact effect of the presence of isolation piles on the form of the velocity field, there is no doubt that the stiffness value of the isolation piles is inversely proportional to the degree of influence. As the stiffness continually tends to infinity, stress concentrations will occur at the base of the pile due to the rigid restraint, resulting in the velocity fields that may move ever closer to the base of the pile.

In the future, more in-depth quantitative studies on pile-soil interaction and more real deformation mechanisms should be developed so that researchers can obtain accurate upper bound solutions.

## Supporting information

**S1 Fig. Relationships of $N_c$ with $e$ and $t$ ($H/B$ = 1.0).**
(OGGU)

**S2 Fig. Relationships of $N_c$ with $e$ and $t$ ($H/B$ = 4.0).**
(OGGU)

**S3 Fig. Relationship between stability coefficient and $t$ ($e$ = 1m).**
(OGGU)

**S4 Fig. Relationship between stability coefficient and $t$ ($e$ = 7m).**
(OGGU)

**S5 Fig. Relationship between the stability coefficient and $e$ ($t$ = 9m).**
(OGGU)

**S6 Fig. Relationship between the stability coefficient and $e$ ($t$ = 15m).**
(OGGU)

**S1 Table. Relationships of $N_c$ with $e$ and $t$ ($H/B$ = 1.0).**
(OGWU)

**S2 Table. Relationships of $N_c$ with $e$ and $t$ ($H/B$ = 4.0).**
(OGWU)

**S3 Table. Relationship between stability coefficient and $t$ ($e$ = 1m).**
(OGWU)

**S4 Table. Relationship between stability coefficient and $t$ ($e$ = 7m).**
(OGWU)

**S5 Table. Relationship between the stability coefficient and $e$ ($t$ = 9m).**
(OGWU)

**S6 Table. Relationship between the stability coefficient and $e$ ($t$ = 15m).**
(OGWU)

**S7 Table. The stability coefficient with no isolation piles.**
(OGWU)

## Author Contributions

**Conceptualization:** Yi Zhou.

**Formal analysis:** Yi Zhou.

**Funding acquisition:** Shaokun Ma.

**Investigation:** Shaokun Ma.

**Methodology:** Yi Zhou.

**Project administration:** Shaokun Ma.

**Resources:** Yu Shao.

**Software:** Yi Zhou.

**Supervision:** Yu Shao, Shaokun Ma.

**Visualization:** Yu Shao.

**Writing – original draft:** Yi Zhou.

**Writing – review & editing:** Shaokun Ma.

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
