## [Decision Letter · Decision Letter 0]

3 Feb 2023

PONE-D-22-32141Analysis of the effects of isolation piles on the basal stability of foundation pits against upheaval based on continuous velocity fieldsPLOS ONE

Dear Dr. Ma,

Thank you for submitting your manuscript to PLOS ONE. After careful consideration, we feel that it has merit but does not fully meet PLOS ONE’s publication criteria as it currently stands. Therefore, we invite you to submit a revised version of the manuscript that addresses the points raised during the review process.

We look forward to receiving your revised manuscript.

Kind regards,

Muhammad Usman

Academic Editor

PLOS ONE

Journal Requirements:

"The project is funded by the National Natural Science Foundation of China (No.52268062), Guangxi Natural Science Foundation Key Project (No.2020GXNSFDA238024), Key Science and Technology Projects in Guangxi’s Transport industry (No.GXJT-2020-02-08), Guangxi Key R&D Program Project (No.GUIKE2021AB22177)."

Reviewers' comments:

Reviewer's Responses to Questions

**Comments to the Author**

1. Is the manuscript technically sound, and do the data support the conclusions?

Reviewer #1: Yes

Reviewer #2: Yes

2. Has the statistical analysis been performed appropriately and rigorously? 

Reviewer #1: Yes

Reviewer #2: N/A

3. Have the authors made all data underlying the findings in their manuscript fully available?

Reviewer #1: Yes

Reviewer #2: Yes

4. Is the manuscript presented in an intelligible fashion and written in standard English?

Reviewer #1: Yes

Reviewer #2: Yes

5. Review Comments to the Author

Reviewer #1: Manuscript Number: PONE-D-22-32141

Manuscript Title: Analysis of the effects of isolation piles on the basal stability of foundation pits against upheaval based on continuous velocity fields

PLOS ONE

The paper presents an interesting subject related to the behavior of isolation piles. The following notes were outlined:

1. Page 1: The “Introduction” must start with a definition of “isolation piles”.

2. Page 2: The following studies may be beneficial. You can also refer to them:

• Al-Shakarchi, Y. J., Fattah, M. Y., and Al-Hadidi, M. T., (2003). “Stability of Unreinforced and Reinforced Embankments on Soft Soils”, Proceedings of the Fifth Scientific Conference of the College of Engineering, University of Baghdad, p.p. 147 – 161.

• Al-Juari, K, A. K., Fattah, M. Y., Khattab, S. I. A., Al-Shamam, M. K., (2020), “Simulation of Behaviour of Swelling Soil Supported by a Retaining Wall”, Proceedings of the Institution of Civil Engineers – Structures and Buildings, https://doi.org/10.1680/jstbu.19.00152.

3. Page 4: What is the meaning o saturated unit weight multiplied by the vertical velocity ?.

4. Page 8: “When the pile length exceeds the excavation depth, the calculation error of the stability coefficient gradually increases when pile-pit spacing remains unchanged.” Why?

5. The assumptions and limitations of study must be mentioned in the beginning of the “Conclusions”.

Reviewer #2: 1. Please explain briefly the method you have adopted in calculating the basal stability of foundation pits against upheaval considering the supporting capacity of isolation piles in last paragraph of “Section 1. Introduction”.

2. Please cite more references in support of your argument that isolation piles can effectively limit soil displacement. A suggestion is the following study

“Mechanism of Isolating Piles in Reducing Tunnel Settlement of Hong Kong-Zhuhai-Macao Bridge Project”

3. Please comment if the authors have produced Table 1 themselves or have taken from some standard. If latter is true, please cite the source.

4. Please cite the source of Table 2.

5. Please correct t is description of Eq. (18).

6. Although the authors have not considered the effects of isolation piles on the form of velocity fields, but they should comment on the impact of this effect on the continuous velocity field considered.

7. Please cite reference of the method “shear strength reduction finite element method (SSRFEM)” mentioned in “Section 5. Numerical Simulation”.

8. The details of numerical model developed and numerical tool used are not provided in “Section 5. Numerical Simulation”.

9. Please comment on the discrepancies seen in the validation of numerical results with the theoretical results in Fig 2.

10. Please consult a native English speaker for proof reading the manuscript.

6. PLOS authors have the option to publish the peer review history of their article (what does this mean?). If published, this will include your full peer review and any attached files.

Reviewer #1: **Yes: **Mohammed Y Fattah

Reviewer #2: No

---

## [Author Response · Author response to Decision Letter 0]

1 Mar 2023

We would like to thank the academic editors and reviewers for their comments, to which we have responded below.

Response to Academic Editors

Response: We have made the required changes.

2. Please state what role the funders took in the study. If the funders had no role, please state: "The funders had no role in study design, data collection, and analysis, decision to publish, or preparation of the manuscript." If this statement is not correct you must amend it as needed. Please include this amended Role of Funder statement in your cover letter; we will change the online submission form on your behalf.

Response: We have added the role of the funders in this study to the cover letter.

3. In your Data Availability statement, you have not specified where the minimal data set underlying the results described in your manuscript can be found. PLOS defines a study's minimal data set as the underlying data used to reach the conclusions drawn in the manuscript and any additional data required to replicate the reported study findings in their entirety. 

Response: We have updated the data availability statement in the cover letter. The raw/processed data required to reproduce these findings cannot be shared at this time as the data also forms part of an ongoing study. 

4. PLOS requires an ORCID iD for the corresponding author in Editorial Manager on papers submitted after December 6th, 2016. Please ensure that you have an ORCID iD and that it is validated in Editorial Manager. 

Response: The corresponding author has linked an ORCID iD to the Editorial Manager account.

Response: We have confirmed the completeness and accuracy of the reference list.

Response to Reviewer #1

1. Page 1: The “Introduction” must start with a definition of “isolation piles”.

Response: We have made the required changes.

2. Page 2: The following studies may be beneficial. You can also refer to them:

• Al-Shakarchi, Y. J., Fattah, M. Y., and Al-Hadidi, M. T., (2003). “Stability of Unreinforced and Reinforced Embankments on Soft Soils”, Proceedings of the Fifth Scientific Conference of the College of Engineering, University of Baghdad, p.p. 147 – 161.

• Al-Juari, K, A. K., Fattah, M. Y., Khattab, S. I. A., Al-Shamam, M. K., (2020), “Simulation of Behaviour of Swelling Soil Supported by a Retaining Wall”, Proceedings of the Institution of Civil Engineers – Structures and Buildings, https://doi.org/10.1680/jstbu.19.00152.

Response: We have added these helpful references.

3. Page 4: What is the meaning o saturated unit weight multiplied by the vertical velocity?

Response: It represents the power of the saturated soil to do work by its own gravity which is a simplified representation of Eq. (1) after a series of derivative calculations.

4. Page 8: “When the pile length exceeds the excavation depth, the calculation error of the stability coefficient gradually increases when pile-pit spacing remains unchanged.” Why?

Response: In the analysis that follows, we determine that this is because in the assumptions of this paper, when the pile length exceeds the excavation depth, the part beyond excavation depth will experience a soil arching effect. It will subject soil movement to more restrictions and cause the calculation error of long piles greater than the actual value. As the results are obtained by integration, the above errors will gradually accumulate with increasing pile lengths, leading to a decrease in the calculation accuracy of long piles.

5. The assumptions and limitations of study must be mentioned in the beginning of the “Conclusions”.

Response: We have added a description of the specific assumptions used in this study, in the section discussing the shortcomings of this study.

Response to Reviewer #2

1. Please explain briefly the method you have adopted in calculating the basal stability of foundation pits against upheaval considering the supporting capacity of isolation piles in last paragraph of “Section 1. Introduction”.

Response: We have made the required changes.

2. Please cite more references in support of your argument that isolation piles can effectively limit soil displacement. A suggestion is the following study

“Mechanism of Isolating Piles in Reducing Tunnel Settlement of Hong Kong-Zhuhai-Macao Bridge Project”

Response: We have added this helpful reference.

3. Please comment if the authors have produced Table 1 themselves or have taken from some standard. If latter is true, please cite the source.

Response: We have added the sources from Table 1.

4. Please cite the source of Table 2.

Response: We have added the sources from Table 2.

5. Please correct t is description of Eq. (18).

Response: We have made the correction.

6. Although the authors have not considered the effects of isolation piles on the form of velocity fields, but they should comment on the impact of this effect on the continuous velocity field considered.

Response: We have added some conjecture on the effect of isolation piles on the velocity field in the concluding discussion section. Obviously, these conjectures lack sufficient evidence, but we still hope to address this issue in future studies, which will be the direction of our subsequent work.

7. Please cite reference of the method “shear strength reduction finite element method (SSRFEM)” mentioned in “Section 5. Numerical Simulation”.

Response: We have added citations to the beginning of the paragraph.

8. The details of numerical model developed and numerical tool used are not provided in “Section 5. Numerical Simulation”.

Response: We have added the main software used in the research process at the beginning of the section.

9. Please comment on the discrepancies seen in the validation of numerical results with the theoretical results in Fig 2.

Response: We have added a comment about the discrepancies seen in the validation of numerical results with the theoretical results in Fig 2.

10. Please consult a native English speaker for proof reading the manuscript.

Response: We have asked a native English speaker to proofread the manuscript and correct some writing errors.

---

## [Editor Report · Decision Letter 1]

14 Mar 2023

Analysis of the effects of isolation piles on the basal stability of foundation pits against upheaval based on continuous velocity fields

PONE-D-22-32141R1

Dear Dr. Ma,

We’re pleased to inform you that your manuscript has been judged scientifically suitable for publication and will be formally accepted for publication once it meets all outstanding technical requirements.

Kind regards,

Muhammad Usman

Academic Editor

PLOS ONE
---

## [Editor Report · Acceptance letter]

10 Apr 2023

PONE-D-22-32141R1 

Analysis of the effects of isolation piles on the basal stability of foundation pits against upheaval based on continuous velocity fields 

Dear Dr. Ma:

I'm pleased to inform you that your manuscript has been deemed suitable for publication in PLOS ONE. Congratulations! Your manuscript is now with our production department. 

Kind regards, 

on behalf of

Dr. Muhammad Usman 

Academic Editor

PLOS ONE